# LeGrad:

# An Explainability Method for Vision Transformers via Feature Formation Sensitivity

## Abstract

Vision Transformers (ViTs) have become a standard architecture in computer vision. However, because of their modeling of long-range dependencies through self-attention mechanisms, the explainability of these models remains a challenge. To address this, we propose LeGrad, an explainability method specifically designed for ViTs. LeGrad computes the gradient with respect to the attention maps of single ViT layers, considering the gradient itself as the explainability signal. We aggregate the signal over all layers, combining the activations of the last as well as intermediate tokens to produce the merged explainability map. This makes LeGrad a conceptually simple and an easy-to-implement method to enhance the transparency of ViTs. We evaluate LeGrad in various setups, including segmentation, perturbation, and open-vocabulary settings, showcasing its improved spatial fidelity as well as its versatility compared to other SotA explainability methods.

## 1 Introduction

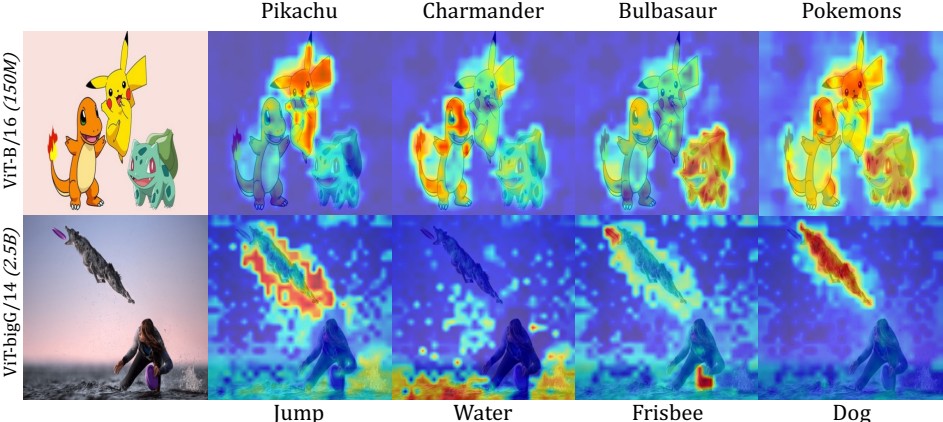

Figure 1: **LeGrad explainability maps:** For a given vision-language model and a textual prompt, LeGrad generates a heatmap indicating the part of the image that is most sensitive to that prompt. Examples shown for OpenCLIP ViT-B/16(150M params.) and ViT-bigG/14(2B params.).

Vision Transformers (ViTs)(Dosovitskiy et al., 2020) have significantly influenced the field of computer vision with their ability to model long-range dependencies through self-attention mechanisms. But explanability methods designed for convolutional or feed-forward neural networks are not directly applicable to ViTs due to their architectural requirements, like GradCAM's (Selvaraju et al., 2017) reliance on convolutional layers and Layer-wise Relevance Propagation's (LRP) (Bach et al., 2015) specific layer-wise propagation rules. While ViT-specific explainability techniques exist, including adaptations of traditional methods (Chefer et al., 2020; Selvaraju et al., 2017; Chefer et al., 2021), attention-based techniques (Abnar & Zuidema, 2020; Voita et al., 2019; Chefer et al., 2020;

2021), and text-based explanations (Hernandez et al., 2021; Goh et al., 2021; Abnar & Zuidema, 2020), the explainability those architectures remains a challenge.

To address this problem, we propose **LeGrad**, a **L**ayerwise **E**xplainability method that considers the **Grad**ient with respect to the attention maps. LeGrad is specifically designed for ViTs as it leverages the self-attention mechanism to generate relevancy maps highlighting the most influential parts of an image for the model's prediction. Compared to other methods, LeGrad uses the gradient with respect to the attention maps as the explanatory signal, as e.g. opposed to CheferCam (Chefer et al., 2020; 2021), which uses the gradient to weight the attention maps. This is done independently for each layer. The final explainability signal is then pooled over all layers of the ViT. Note that using a layerwise gradient, compared to other signals, allows to sum up over different layers without further need for normalization. To further improve the signal, the gradient is clipped by a ReLU function preventing negative gradients to impact positive activations (see Figure 2 for details). The approach is conceptually simple and versatile, as it only requires the gradient w.r.t. to the ViT's attention maps. This facilitates its adoption across various applications and architecture, including larger ViTs such as ViT-BigG as well as attention pooling architectures (Lee et al., 2019; Zhai et al., 2023).

We evaluate the proposed method for various ViT backbones on four challenging tasks, segmentation, open-vocabulary detection, perturbation, and audio localization, spanning over various datasets, incl. ImageNet (Russakovsky et al., 2015; Gao et al., 2022), OpenImagesV7 (Benenson & Ferrari, 2022), and ADE20KSound/SpeechPrompted (Hamilton et al., 2024) . It shows that while current methods struggle especially with the diverse object categories in OpenImagesV7, LeGrad reaches a score of $48.4$ *p-mIoU* on OpenImagesV7 using OpenCLIP-ViT-B/16. Furthermore, we demonstrate the applicability of LeGrad to very large models, such as the ViT-BigG/14 (Cherti et al., 2023) with 2.5 billion parameters while also adapting well to different feature aggregation strategies employed e.g. by SigLIP (Zhai et al., 2023). Finally, LeGrad also establishes a new SoTA on zero-shot sound localization on ADE20KSoundPrompted scoring $+14mIoU$ over previous SoTA.

We summarize the contributions as follows: (1) We propose LeGrad as a layerwise explainability method based on the gradient with respect to ViTs attention maps. (2) As the layerwise explainability allows to easily pool over many layers, LeGrad scales to large architectures such as ViT-BigG/14 and is applicable to various feature aggregation methods. (3) We evaluate LeGrad on various tasks and benchmarks, showing its improvement compared to other state-of-the-art explainability methods especially for large-scale open vocabulary settings.

## 2 RELATED WORK

**Gradient-Based Explanation Methods** Feature-attribution methods are a commonly used explanation technique that explains model decisions by assigning a score to each image pixel, representing its importance to the model's output. Generally, these methods can be categorized into two groups (Molnar, 2019) — gradient-based methods that compute explanations based on the gradient of the prediction of the model with respect to each input pixel (Simonyan et al., 2014; Erhan et al., 2009; Sundararajan et al., 2017; Springenberg et al., 2015; Smilkov et al., 2017; Kapishnikov et al., 2019; Selvaraju et al., 2017) and perturbation-based methods that measure pixel importance by successively perturbing the input images and measuring the impact on the model output (Lundberg & Lee, 2017; Ribeiro et al., 2016; Petsiuk et al., 2018; Carter et al., 2019; Zeiler & Fergus, 2014). While both types of methods have been used successfully to identify correlations and trustworthiness in traditional computer vision models (Boggust et al., 2022; Carter et al., 2021), gradient-based methods are often more computationally efficient since they only require a single backwards pass. Further, they are easy to interpret since they are a direct function of the model's parameters and do not rely on additional models or image modifications. However, many existing gradient-based methods were designed for convolutional and feed-forward model architectures, so it is non-trivial to directly apply them to ViTs since ViTs do not contain spatial feature maps and include complex interactions between patches induced by the self-attention mechanism. As most gradient-based methods were designed prior to the widespread use of ViTs, researchers have recently made efforts to adapt existing methods and to develop new ones specifically for transformers. Chefer et al. (Chefer et al., 2020) extend LRP (Bach et al., 2015) to transformers by integrating gradients within the self-attention layers. However, this approach is computationally heavy and is inflexible to architecture changes as it requires specific implementation for each module of the network. To

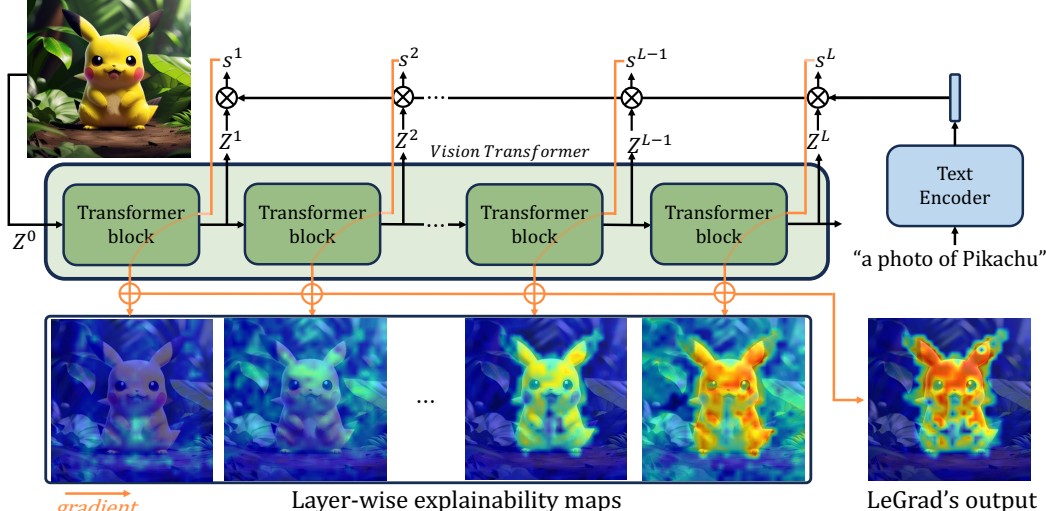

$\xrightarrow{gradient}$       Layer-wise explainability maps       LeGrad's output

Figure 2: **Overview of LeGrad:** Given a text prompt or a classifier $\mathcal{C}$, an activation $s^l$ is computed for each layer $l$. The activation $s^l$ is then used to compute the explainability map of that layer. The layerwise explainability maps are then merged to produce LeGrad's output.

circumvent that complexity, CheferCAM (Chefer et al., 2021) weights the attention by their gradient and aggregates it through the layer via matrix multiplication. However, the use of gradients to weigh the attention heads' importance makes this method class-specific.

**Explanability Methods for ViT** A separate line of research has proposed using ViT attention maps, as opposed to gradients, as a way to explain for transformers' decisions (Abnar & Zuidema, 2020; Voita et al., 2019). One attention-based method, rollout (Abnar & Zuidema, 2020), traces the flow of importance through the transformer's layers by linearly combining the attention maps via matrix multiplication. Attention flow (Abnar & Zuidema, 2020) contextualizes the attention mechanism as a max-flow problem; however, it is computationally demanding and has not been extensively evaluated for vision tasks. While these methods offer insights into the attention mechanism, they often neglect the non-linear interactions between attention heads and the subsequent layers. Moreover, they may not adequately distinguish between positive and negative contributions to the final decision, leading to potentially misleading interpretations, as found in Chefer et al. (Chefer et al., 2020). Compared to that LeGrad uses the gradient w.r.t. to the attention maps, thereby assessing the sensitivity of the attention maps to a change in the patch tokens.

**Vision-Language Explainability Methods** Research has also explored vision-langauge models for interpreting representations in vision models. For instance, leveraging CLIP's language-image space, researchers have provided text descriptions for active neuron regions (Hernandez et al., 2021; Goh et al., 2021) and projected model features into text-based concept banks (Gandelsman et al., 2023). In particular, TEXTSPAN (Gandelsman et al., 2023) focuses on the explainability of CLIP-like models. It refrains from using gradient computation by aggregating the intermediate features' similarities along a given text direction, creating a relevancy map for a text query. LeGrad advances this line of work by focusing on the sensitivity of feature representations within ViTs to generate relevancy maps that can be adapted to various feature aggregation strategies.

## 3 METHOD

In this section, we first introduce ViT's mechanics and the different feature aggregation mechanisms used for this architecture. We then explain the details of LeGrad, starting by a single layer and then extending it to multiple layers.

### 3.1 BACKGROUND: FEATURE FORMATION IN ViTs

The ViT architecture is a sequence-based model that processes images by dividing them into a grid of $n$ patches. These patches are linearly embedded and concatenated with a class token $z_0^0 = z_{[CLS]}^0 \in \mathbb{R}^d$, which is designed to capture the global image representation for classification tasks. The input

image $I$ is thus represented as a sequence of $n+1$ tokens $Z^0 = \{z_0^0, z_1^0, \ldots, z_n^0\}$, each of dimension $d$, with positional encodings added to retain spatial information.

The transformation of the initial sequence $Z^0 \in \mathbb{R}^{(n+1) \times d}$ through the ViT involves $L$ layers, each performing a series of operations. Specifically, each layer $l$ applies multi-head self-attention (MSA) followed by a multilayer perceptron (MLP) block, both with residual connections:

$$\hat{Z}^l = \text{MSA}^l(Z^{l-1}) + Z^{l-1}, \quad Z^l = \text{MLP}^l(\hat{Z}^l) + \hat{Z}^l. \tag{1}$$

After $L$ layers, the image representation can be obtained via various strategies:

**[CLS] token:** The class token approach, as introduced in ViT (Dosovitskiy et al., 2020), uses the processed class token as the image embedding $\bar{z}_{[CLS]} = z_0^L$. This method relies on the transformer's ability to aggregate information from the patch tokens into the class token during training.

**Attentional Pooler:** Attention Pooling, as e.g. used in SigLIP (Zhai et al., 2023) employs an multi-head attention layer (Lee et al., 2019; Yu et al., 2022) with a learnable query token $q_{pool} \in \mathbb{R}^d$. This token interacts with the final layer patch tokens to produce the pooled representation $\bar{z}_{AttnPool}$:

$$\bar{z}_{AttnPool} = \text{softmax}\left(\frac{q_{pool} \cdot (W_K Z^L)^T}{\sqrt{d}}\right)(W_V Z^L), \tag{2}$$

where $W_K, W_V \in \mathbb{R}^{d \times d}$ are learnable projection matrices.

Independent of the feature aggregation strategy, it is important for an explainability method to account for the iterative nature of feature formation in ViTs and to capture the contributions of all layers for the final representation. LeGrad addresses this by fusing information from each layer, allowing for both, a granular as well as a joint, holistic interpretation of the model's predictions.

## 3.2 Explainability Method: LeGrad

We denote the output tokens of each block $l$ of the ViT as $Z^l = \{z_0^l, z_1^l, \ldots, z_n^l\} \in \mathbb{R}^{(n+1) \times d}$, where $d$ is the dimensionality of each token, and $\bar{z}^l$ is the average over the tokens of the respective layer $l$, defined as $\bar{z}^l = \frac{1}{n+1} \sum_{i=0}^n z_i^l$.

Consider a mapping model $\mathcal{C} \in \mathbb{R}^{d \times C}$, that maps the token dimension $d$ to $C$ logits. This classifier can be learned during training for a supervised classification task (e.g., ImageNet) and in that case $C$ is the number of classes. Or it can be formed from text embeddings of some prompts in the case of zero-shot classifier, *e.g.,* vision-language models like CLIP, then $C$ is the number of prompts. For a given layer $l$, the mapping model $\mathcal{C}$ then generates a prediction $\bar{y}^l$, which can be the output of the classifier or, in case of vision-language models, the vector of results of dot products of the text and image embeddings. This prediction $\bar{y}^l$ is obtained by passing the aggregated feature representation of the ViT, noted $\bar{z}^l$, through the mapping $\mathcal{C}$:

$$\bar{y}^l = \bar{z}^l \cdot \mathcal{C} \in \mathbb{R}^C. \tag{3}$$

Note that most explainability methods only use the final outputs of the model. We argue that also leveraging intermediate representations is beneficial (see Section 4.7). The following we first describe how to obtain the explainability map for a single layer, using an arbitrary layer $l$ as an example and then generalize it to multiple layers. The overall method is visualized Figure 2.

**Process for a Single Layer:** To compute a 2D map that highlights the image regions most influential for the model's prediction of a particular class, we focus on the activation with respect to the target class/prompt $\hat{c}$, denoted by $s^l = \bar{y}_{[\hat{c}]}^l$. The attention operation within a ViT is key to information sharing, and thus our method concentrates on this process.

We compute the gradient of the activation $s^l$ with respect to the attention map of layer $l$, as shown in Figure 3, denoted as $\mathbf{A}^l \in \mathbb{R}^{h \times n \times n}$:

$$\nabla \mathbf{A}^l = \frac{\partial s}{\partial \mathbf{A}^l} \in \mathbb{R}^{h \times (n+1) \times (n+1)}, \tag{4}$$

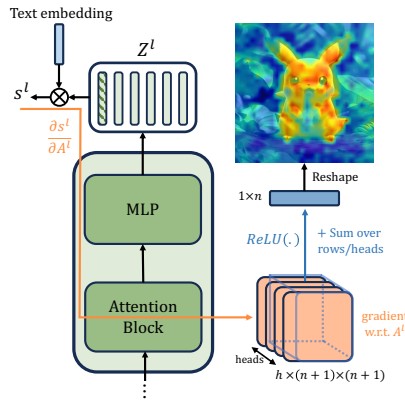

Figure 3: **LeGrad** for a single layer.

where $h$ is the number of heads in the self-attention operation. Negative gradients are discarded by a ReLU function (noted $(.)^+$), and the gradient is averaged across the patch and head dimensions:

$$\hat{E}^l(s) = \frac{1}{h \cdot (n+1)} \sum_h \sum_i \left(\nabla \mathbf{A}_{h,i,.}^l\right)^+ \in \mathbb{R}^{n+1}. \tag{5}$$

To obtain the final explainability map, the column corresponding to the [CLS] token is removed, only considering the patch tokens, reshaped into a 2D map, and a min-max normalization is applied:

$$E^l(s) = \text{norm}(\text{reshape}(\hat{E}^l(s)_{1:})) \in \mathbb{R}^{W \times H}. \tag{6}$$

**Process for Multiple Layers:** Recognizing that information aggregation occurs over several layers, we extend the process to all layers. For each layer $l$, we calculate the activation score $s^l$ using the intermediate tokens $Z^l$ and derive the explainability map accordingly:

$$\hat{E}^l(s^l) = \frac{1}{h \cdot (n+1)} \sum_h \sum_i \left(\nabla \mathbf{A}_{h,i,.}^l\right)^+ \in \mathbb{R}^{n+1}. \tag{7}$$

We then average the explainability maps from each layer:

$$\bar{\mathbf{E}} = \frac{1}{L} \sum_l \hat{E}^l(s^l)_{1:} \tag{8}$$

And finally we reshape to the original image size and apply min-max normalization:

$$\mathbf{E} = \text{norm}(\text{reshape}(\bar{\mathbf{E}})) \in R^{W \times H}. \tag{9}$$

This queries each patch token at a given layer about its influence on the prediction at that stage.

**Adaptation to Attentional Pooler:** For ViTs using an attentional pooler (*e.g.* SigLIP (Zhai et al., 2023)), a slight modification is made to compute the activation $s^l$ at each layer. We apply the attentional pooler module $Attn_{pool}$ to each intermediate representation $Z^l$ to obtain a pooled query $q^l \in \mathbb{R}^d$. The activation $s^l$ with respect to the desired class $c$ is then computed as $s^l = q^l \cdot \mathcal{C}_{:,c} \in \mathbb{R}$. Instead of considering the self-attention map, we use the attention map of the attentional pooler, denoted $\mathbf{A}_{pool} \in \mathbb{R}^{h \times 1 \times n}$. Thus, for every layer $l$, $\nabla A^l = \frac{\partial s^l}{\partial A_{pool}^l}$.

## 4 EXPERIMENTS

### 4.1 OBJECT SEGMENTATION

Following standard benchmarks (Chefer et al., 2020; 2021; Gandelsman et al., 2023) we evaluate the ability of explainability methods to accurately localize an object in the image. ◇***Task:*** To do so, we generate image heatmaps based on the activation of the groundtruth class for models trained with a classifier or based on the the class descriptions *"A photo of a [class]"* for vision-language models. Subsequently, we apply a threshold to binarize these heatmaps (using a threshold of 0.5), thereby obtaining a foreground/background segmentation. ◇***Metric:*** We assess the quality of this segmentation by computing the mIoU (mean Intersection over Union), pixel accuracy and the mAP (mean Average Precision) zero-shot segmentations produced by different explainability methods. This benchmark serves as a testbed for evaluating the spatial fidelity of the explainability method. ◇**Dataset:** In our evaluation of heatmap-based explainability methods, we adhere to a standardized protocol and use the ImageNet-Segmentation dataset (Gao et al., 2022), with $4,276$ images that provide segmentation annotations.

Table 1 compares LeGrad as well as other methods in the context of image segmentation using the ImageNet-segmentation dataset. LeGrad achieved a mIoU of 58.7%, surpassing other SOTA explainability methods. Notably, it outperformed CheferCAM as a gradient-based method for ViTs, and TextSpan as a non-gradient-based method, indicating its robustness in capturing relevant image features for classification tasks.

| Method | Pixel Acc.↑ | mIoU↑ | mAP↑ |
|---|---|---|---|
| LRP | 52.81 | 33.57 | 54.37 |
| Partial-LRP | 61.49 | 40.71 | 72.29 |
| rollout | 60.63 | 40.64 | 74.47 |
| Raw attention | 65.67 | 43.83 | 76.05 |
| GradCAM | 70.27 | 44.50 | 70.30 |
| CheferCAM | 69.21 | 47.47 | 78.29 |
| TextSpan | 73.01 | 40.26 | 81.4 |
| LeGrad | **77.52** | **58.66** | **82.49** |

Table 1: **Object Segmentation:** method comparison on ImageNet-S using an OpenCLIP(ViT-B/16) model trained on Laion2B.

| Method | p-mIoU ↑ | | |
|---|---|---|---|
| | B/16 | L/14 | H/14 |
| rollout | 8.75 | 6.85 | 5.82 |
| Raw attention | 0.94 | 1.60 | 0.85 |
| GradCAM | 8.72 | 2.80 | 2.46 |
| AttentionCAM | 5.87 | 4.74 | 1.20 |
| CheferCAM | 5.87 | 2.51 | 9.49 |
| TextSpan | 9.44 | 21.73 | 23.74 |
| LeGrad | **48.38** | **47.69** | **46.51** |

Table 2: **Open-Vocabulary Segmentation:** methods comparison on OpenImagesV7 using different OpenCLIP model sizes.

| Method | Speech Seg. | | Sound Seg. | |
|---|---|---|---|---|
| | mAP↑ | mIoU↑ | mAP↑ | mIoU↑ |
| DAVENet | 32.2 | 26.3 | 16.8 | 17.0 |
| CAVMAE | 27.2 | 19.9 | 26.0 | 20.5 |
| ImageBind | 20.2 | 19.7 | 18.3 | 18.1 |
| ImageBind + LeGrad | **23.3** | **21.8** | **48.0** | **38.9** |
| DenseAV* | **48.7** | **36.8** | 32.7 | 24.2 |

Table 3: **Speech and Sound prompted semantic segmentation:** Comparison of the sound localization methods on ADE20K Speech & Sound Prompted dataset (Hamilton et al., 2024).

| Method | fps |
|---|---|
| AttentionCAM | 103 |
| GradCAM | 108 |
| CheferCAM | 21 |
| LRP | 4.0 |
| TextSpan | 3.8 |
| LeGrad | 96 |

Table 4: Speed comparison for ViT-B/16

### 4.2 OPEN-VOCABULARY LOCALIZATION

For vision-language models, we extend our evaluation to encompass open-vocabulary scenarios by generating explainability maps for arbitrary text descriptions. This allows us to assess the quality of explainability methods beyond the common classes found in ImageNet. ◇*Task:* We generate a heatmap for each class object present in the image, binarize them (using a threshold of 0.5) and assess the localization accuracy. ◇**Dataset/Metric:** We employ the OpenImageV7 dataset (Benenson & Ferrari, 2022), which offers annotations for a diverse array of images depicting a broad spectrum of objects and scenarios. Following (Bousselham et al., 2023), our evaluation utilizes the point-wise annotations of the validation set, which contains 36,702 images labeled with 5,827 unique class labels. Each image is associated with both positive and negative point annotations for the objects present. In our analysis, we focus exclusively on the classes that are actually depicted in each image.

Table 2 evaluates the performance on the OpenImagesV7 dataset testing the capabilities of all methods in handling diverse object categories. Note that for GradCAM we searched over the layers and took the one that was performing the best. LeGrad outperforms all other SOTA methods, with performance gains ranging from $2\times$ to $5\times$ compared to the second-best performing method. This can be seen as an indicator for LeGrad's capacity for fine-grained recognition.

### 4.3 AUDIO LOCALIZATION

To further validate LeGrad versatility, we measure its ability to be used with audio-visual models. We use ImageBind(Girdhar et al., 2023), a model trained to align several modality with the same image encoder, in particular the audio modality. We choose this model as it is widely used and the image encoder is a vanilla ViT, hence compatible with LeGrad. ◇*Task:* Given an audio prompt, we generate a heatmap that localize the part of the image that correspond to that audio. ◇*Dataset/Metric:* We use the recently proposed ADE20kSoundPrompted and ADE20kSpeechPrompted(Hamilton et al., 2024) to measure the audio localization of ImageBind + LeGrad. Following (Hamilton et al., 2024), we report the mean Average Precision (*mAP*) and the mean Intersection over Union (*mIoU*) computed using the ground truth masks.

Table 3 compares the sound localization performance of different state-of-the-art methods. We observe that when applied to the ImageBind model (Girdhar et al., 2023), LeGrad leads to a significant

increase in sound segmentation performance, hence establishing a new SOTA on that benchmark by outperforming DAVENet (Harwath et al., 2018), CAVMAE (Gong et al., 2023) and the previous SOTA DenseAV (Hamilton et al., 2024). The less pronounced improvement of LeGrad over ImageBind on speech segmentation is due to the fact that ImageBind was not trained with speech data (only sound). Therefore, since ImageBind performs poorly on speech, LeGrad does not further improve the localization.

## 4.4 PERTURBATION-BASED EVALUATION

Next, to measure LeGrad's ability to faithfully identify features important to the model, we employ a perturbation-based methodology. ◇***Task:*** Given a classification dataset, we begin by generating explainability maps for every image using the different explainability methods. The analysis then consists of two complementary perturbation tests: positive and negative. In the positive perturbation test, image regions are occluded in descending order of their attributed relevance, as indicated by the explainability maps. Conversely, the negative perturbation test occludes regions in ascending order of relevance (see the *Annex* for more details and visualizations of positive/negative perturbations). ◇***Metric:*** For both perturbation scenarios, we quantify the impact on the model's accuracy by computing the area under the curve (AUC) for pixel erasure, which ranges from 0% to 90%. This metric provides insight into the relationship between the relevance of image regions and the model's performance. The tests are applicable to both the predicted and ground-truth classes, with the expectation that class-specific methods will show improved performance in the latter. This dual perturbation approach enables a comprehensive evaluation of the network's explainability by highlighting the importance of specific image regions in the context of the model's classification accuracy. ◇**Dataset:** Following common practices, we use the ImageNet validation set, which contains $50K$ images and covers $1,000$ classes.

As detailed in Table 5, LeGrad's performance is here comparable to TextSpan for positive perturbations and slightly superior for negative perturbations. For all other methods, LeGrad outperformes both attention-based (e.g., "rollout" and "raw attention") and gradient-based methods (e.g., Grad-CAM, AttentionCAM, and CheferCAM) across various model sizes and for both predicted and ground truth classes, emphasizing its ability to identify and preserve critical image regions for accurate classification.

## 4.5 SPEED COMPARISON

Table 4 compares the inference speed for different methods averaged over $1,000$ images. Generally, gradient-based methods are faster than methods like LRP and TextSpan. More specifically, despite using several layers for its prediction, LeGrads speed only drops slightly compared to GradCAM and AttentionCAM, which both use a single layer and are significantly faster than CheferCAM. The observed speed difference stems from summing the contribution of each layer rather than using complex matrix multiplication as in CheferCAM.

## 4.6 PERFORMANCE ON SIGLIP

We also evaluate the adaptability regarding the performance on SigLIP-B/16, a Vision-Language model employing an attentional pooler as shown in Table 6. The results underscore the methods performance across both negative and positive perturbation-based benchmarks. Notably, in the open-vocabulary benchmark on OpenImagesV7, LeGrad achieved a p-mIoU of 25.4, significantly surpassing GradCAM's 7.0 p-mIoU, the next best method. These findings affirm the versatility of LeGrad, demonstrating its robust applicability to various pooling mechanisms within Vision Transformers. Further details on the methodological adaptations of LeGrad and other evaluated methods for compatibility with SigLIP are provided in the annex.

## 4.7 ABLATION STUDIES

**Layer Accumulation.**To further understand the impact of the number of layers considered in the computation of LeGrad's explainability maps, we investigate this aspect across two distinct benchmarks: a perturbation-based evaluation using the ImageNet validation set and an open-vocabulary

| | Method | Negative Pred. ↑ | Negative Targ. ↑ | Positive Pred. ↓ | Positive Targ. ↓ |
|---|---|---|---|---|---|
| ViT-B/16 | rollout | 44.36 | 44.36 | 23.03 | 23.03 |
| | Raw attention | 46.97 | 46.97 | 20.23 | 20.23 |
| | GradCAM | 32.01 | 45.26 | 36.52 | 22.86 |
| | AttentionCAM | 39.56 | 39.68 | 34.28 | 34.12 |
| | CheferCAM | 47.91 | 49.28 | 18.66 | 17.70 |
| | TextSpan | **50.92** | **52.81** | 15.10 | 14.26 |
| | LeGrad | 50.24 | 52.27 | **15.06** | **13.97** |
| ViT-L/14 | rollout | 40.46 | 40.46 | 29.46 | 29.46 |
| | Raw attention | 47.14 | 47.14 | 23.49 | 23.49 |
| | GradCAM | 45.24 | 47.08 | 23.81 | 22.68 |
| | AttentionCAM | 45.81 | 45.84 | 31.18 | 31.03 |
| | CheferCAM | 49.69 | 50.37 | 20.67 | 20.14 |
| | TextSpan | **53.17** | 54.42 | 16.77 | 16.12 |
| | LeGrad | 53.11 | **54.48** | **15.98** | **15.23** |

| | Method | Negative Pred. ↑ | Negative Targ. ↑ | Positive Pred. ↓ | Positive Targ. ↓ |
|---|---|---|---|---|---|
| ViT-H/14 | rollout | 49.37 | 49.37 | 32.91 | 32.91 |
| | Raw attention | 54.42 | 54.42 | 29.25 | 29.25 |
| | GradCAM | 45.26 | 45.48 | 40.98 | 40.91 |
| | AttentionCAM | 51.62 | 51.65 | 36.72 | 36.56 |
| | CheferCAM | 56.55 | 56.56 | 26.17 | 26.16 |
| | TextSpan | **60.14** | **61.91** | 20.07 | 19.14 |
| | LeGrad | 60.02 | 61.72 | **19.30** | **18.26** |
| ViT-BigG/14 | rollout | 38.44 | 38.44 | 48.72 | 48.72 |
| | Raw attention | 56.56 | 56.56 | 31.25 | 31.25 |
| | GradCAM | 40.06 | 40.98 | 57.53 | 56.31 |
| | AttentionCAM | 54.68 | 54.74 | 39.28 | 39.27 |
| | CheferCAM | 57.95 | 58.27 | 29.21 | 28.45 |
| | TextSpan | **63.32** | **64.95** | 21.96 | 21.06 |
| | LeGrad | 62.62 | 64.67 | **21.33** | **20.28** |

Table 5: **SOTA Perturbation Performance:** Comparison of explainability methods on the ImageNet-val using a different model size.

| | ImageNet | | | | OpenImagesV7 |
|---|---|---|---|---|---|
| | Negative | | Positive | | |
| Method | Predicted ↑ | Target ↑ | Predicted ↓ | Target ↓ | p-mIoU ↑ |
| rollout | 47.81 | 47.81 | 25.74 | 25.74 | 0.07 |
| Raw attention | 44.42 | 44.42 | 25.85 | 25.85 | 0.09 |
| GradCAM | 41.25 | 44.42 | 35.10 | 33.50 | 6.97 |
| AttentionCAM | 45.62 | 45.71 | 45.01 | 44.92 | 0.19 |
| CheferCAM | 47.12 | 49.13 | 22.35 | 21.15 | 1.94 |
| LeGrad | **50.08** | **51.67** | **18.48** | **17.55** | **25.40** |

Table 6: **SOTA comparison on SigLIP-B/16:** Comparison of explainability methods on perturbation-based tasks on ImageNet-val and open-vocabulary localization on OpenImagesV7.

segmentation task on the OpenImagesV7 dataset. The experiments are performed on various model sizes from the OpenCLIP library, including ViT-B/16, ViT-L/14, and ViT-H/14.

◇ **Perturbation-Based Evaluation:** In the perturbation-based evaluation (Figure 4 left) we employ a negative perturbation test as described in Section 4.4, using the ground-truth class for reference. The results indicate that the ViT-B/16 model's performance is optimal when fewer layers are included in the explainability map computation. Conversely, larger models such as ViT-L/14 and ViT-H/14 show improved performance with the inclusion of more layers. This suggests that in larger models, the aggregation of information into the [CLS] token is distributed across a greater number of layers, necessitating a more comprehensive layer-wise analysis for accurate explainability.

◇ **Open-Vocabulary Segmentation:** For the open-vocabulary segmentation task (Figure 4 right) all models demonstrate enhanced performance with the inclusion of additional layers in the explainability map computation. The optimal number of layers varies with the size of the model, with larger models requiring a larger number of layers. This finding is consistent with existing literature (Gandelsman et al., 2023), suggesting that the information aggregation process in ViTs is more distributed in larger architectures. However, it is also observed that beyond a certain number of layers, the performance plateaus, indicating that the inclusion of additional layers does not further enhance the explainability map's quality.

To further analyze those effects, we provide a qualitative visualization of the per-layer heatmaps produced by LeGrad in Figure 5. It shows that the localization of the prompt is not confined to a single layer but is distributed across multiple layers. This observation is consistent with the findings from the above ablations, which underscore the necessity of incorporating multiple layers into the explainability framework to capture the full scope of the model's decision-making process.

This observation not only corroborates the utility of incorporating multiple layers into the explainability analysis but also suggests a more distributed information aggregation process into the [CLS] token in larger models, as posited in the literature (Gandelsman et al., 2023).

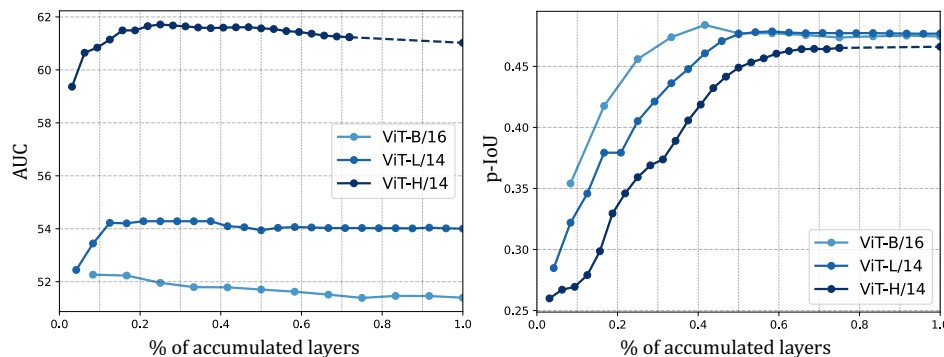

Figure 4: Ablation on the number of layers used in LeGrad for different architecture sizes. Left: AUC for Negative perturbation on ImageNet-val for different layers used for LeGrad. Right: point-mIoU on OpenImagesV7 for different layers used for LeGrad.

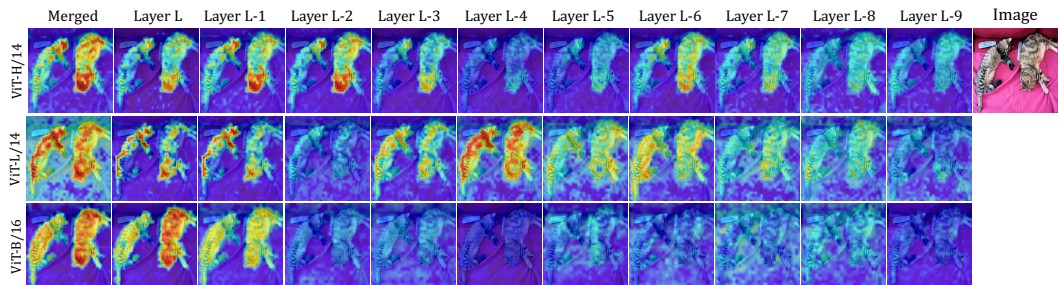

Figure 5: Qualitative analysis of the impact of each layer for different model sizes using *"a photo of a cat"* as prompt. In smaller models the explainability signal predominantly emanates from the final layers while in larger models lower layers also contribute to the explainability map.

**Gradient Distribution Analysis.** To better understand the inner workings of the proposed LeGrad method, we further analyse the gradient intensity distribution across the layers of ViT-L/14 models from OpenCLIP based on Laion400M, OpenAI CLIP, and MetaCLIP. This analysis was performed using the PascalVOC dataset. The explainability maps for each layer were computed in accordance with Equation 9. Subsequently, the associated segmentation mask was utilized to compute the average gradient within the mask for each layer. This process yielded a single value per layer. To account for the impact of the mask size, we apply min-max normalization across all layers. This normalization process ensured that the layer contributing the most to the final explainability map was assigned a value of one, while the least contributing layer was assigned a value of zero. Figure 6 depicts the gradient distribution over the layers for different sets of pretrained ViT-L/14 models. First, we observe that for most models, the layers contributing the most are typically located towards the end of the ViT. Second, despite all sharing the exact same model architecture and being trained with the same loss, we observe significant difference in layer importance. For Laion400M and OpenAI the most important layer is the last one whereas for MetaCLIP it is rather the penultimate. Interestingly, for the Laion400M variant, the middle layers have a more pronounced influence. Overall, this provides a generalized view of the model's behavior, serving as a sort of "fingerprint" for the model. Annex D provides an analysis for more model sizes and weights, as well as a visualization of the gradient distribution over all layers independently for each class in PascalVOC.

## 4.8    QUALITATIVE COMPARISON TO SOTA

Here, we present a qualitative analysis of the explainability maps generated by LeGrad in comparison to other state-of-the-art (SOTA) methods. The visual results are depicted in Figure 7, which includes a diverse set of explainability approaches such as gradient-based methods (e.g., Chefer-CAM, GradCAM), attention-based methods (e.g., Raw Attention weights visualization, Rollout), and methods that integrate intermediate visual representations with text prompts (e.g., TextSpan).

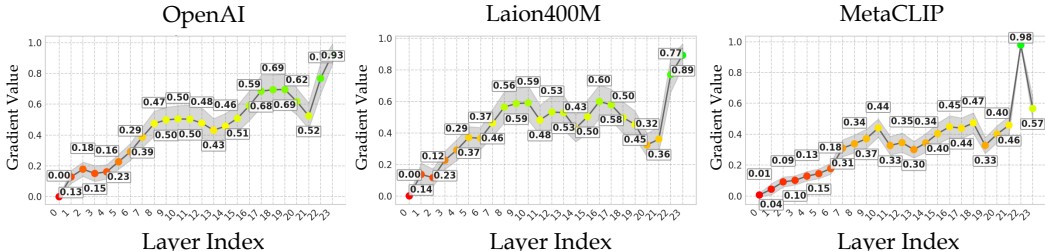

Figure 6: Gradient distribution over layers on PascalVOC for different pretrained ViT-L/14.

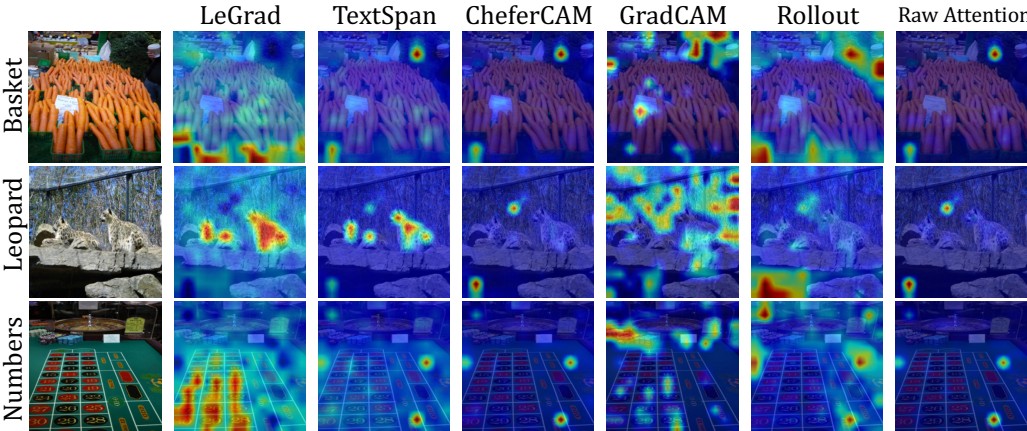

Figure 7: **SOTA Qualitative Comparison:** visual comparison of different explainability methods on images from OpenImagesV7.

Our observations indicate that raw attention visualizations tend to highlight a few specific pixels with high intensity, often associated with the background rather than the object of interest. This pattern, consistent with findings in the literature (Bousselham et al., 2023; Darcet et al., 2023), suggests that certain tokens disproportionately capture attention weights. Consequently, methods that rely on raw attention weights to construct explainability maps, such as CheferCAM, exhibit similar artifacts. For instance, in the localization of "Basket" (Figure 7, row 1), the basket is marginally accentuated amidst a predominance of noisy, irrelevant locations. In contrast, for LeGrad, the presence of uniform noisy activations across different prompts results in minimal gradients for these regions, effectively filtering them out from the final heatmaps. This characteristic enables LeGrad to produce more focused and relevant visual explanations.

## 5 CONCLUSION

In this work, we proposed LeGrad as an explainability method that highlights the decision-making process of Vision Transformers (ViTs) across all layers various feature aggregation strategies. We validated the method's effectiveness in generating interpretable visual explanations that align with the model's reasoning. Our approach offers a granular view of feature formation sensitivity, providing insights into the contribution of individual layers and attention mechanisms. The evaluation across object segmentation, perturbation-based metrics, and open-vocabulary scenarios underscores LeGrad's versatility and fidelity in different contexts. By facilitating a deeper understanding of ViTs LeGrad paves the way for more transparent and interpretable foundation models.

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

## A    OVERVIEW

In this supplementary material, we first provide in section B detailed information regarding the implementation, including pre-trained weights, baseline methodologies, and necessary adaptations for applying GradCAM to Vision Transformers (ViT). An expanded evaluation of LeGrad's performance, specifically on Vision Transformers trained with the ImageNet dataset, is documented in section C. Section C.2 offers a visual representation of the perturbation benchmarks utilized to assess the efficacy of various explainability approaches, along with additional qualitative examples in section C.3 to further illustrate the capabilities of our method. Section D further explores the gradient distribution across layers for different models and datasets. Furthermore, section **??** conducts a sanity check using the FunnyBirds Co-12 setup to evaluate the robustness of our explainability method. Lastly, section F includes a disclaimer addressing the use of personal and human subject data within our research.

## B    IMPLEMENTATION DETAILS

### B.1    PRETRAINED WEIGHTS

The experiments conducted in our study leverage a suite of models with varying capacities, including ViT-B/16, ViT-L/14, ViT-H/14, and ViT-bigG/14. These models are initialized with pre-trained weights from the OpenCLIP library respectively identified by: `"laion2b_s34b_b88k"`, `"laion2b_s32b_b82k"`, `"laion2b_s32b_b79k"`, and `"laion2b_s39b_b160k"`. For the SigLIP method, we utilize the ViT-B/16 model equipped with the `"webl"` weights. For the "gradient distribution over layers" graphs, Figure 6, we also used the pretrained weights from OpenAI (Radford et al., 2021) and MetaCLIP (Xu et al., 2023).

### B.2    DETAILED DESCRIPTION OF BASELINES

In this section, we provide a concise overview of the baseline methods against which our proposed approach is compared.

**GradCAM**: While originally designed for convolutional neural networks (CNNs), GradCAM can be adapted for Vision Transformers (ViTs) by treating the tokens as activations. To compute the GradCAM explainability map for a given activation $s$, we calculate the gradient of $s$ with respect to the token dimensions. The gradients are aggregated across all tokens and serve as weights to quantify the contribution of each token dimension. Formally, for intermediate tokens $Z^l = \{z_0^l, z_1^l, \dots, z_n^l\} \in \mathbb{R}^{(n+1) \times d}$, the GradCAM map $E_{GradCAM}$ is defined as:

$$w = \frac{1}{n} \sum_{i=0}^{n} \frac{\partial s}{\partial z_i^l} \quad \in \mathbb{R}^{d \times 1 \times 1}$$

$$\hat{E}_{GradCAM} = \left( \frac{1}{d} \sum_{k=1}^{d} w_d * Z_{1:,d}^l \right)^{+} \quad \in \mathbb{R}^{n} \tag{10}$$

$$E_{GradCAM} = \text{norm}(\text{resize}(\hat{E}_{GradCAM})) \quad \in \mathbb{R}^{W \times H},$$

with $d$ representing the token dimension, $*$ denoting element-wise multiplication, and the superscript $+$ indicating the ReLU operation. We empirically determined that applying GradCAM to layer 8 of ViT-B/16 yields optimal results.

**AttentionCAM**: This method extends the principles of GradCAM to ViTs by utilizing the attention mechanism within the transformer's architecture. AttentionCAM leverages the gradient signal to weight the attention maps in the self-attention layers. Specifically, for the last block's self-attention maps $\mathbf{A}^L$, the AttentionCAM map $E_{AttnCAM}$ is computed as:

$$\nabla \mathbf{A}^L = \frac{\partial s}{\partial \mathbf{A}^L} \in \mathbb{R}^{h \times (n+1) \times (n+1)}$$

$$w = \frac{1}{n^2} \sum_{i,j} \nabla \mathbf{A}^L_{:,i,j} \in \mathbb{R}^h$$

$$\hat{E}_{AttnCAM} = \sum_p^h \left( w_p * A^L_{p,:,:} \right) \in \mathbb{R}^{(n+1) \times (n+1)} \tag{11}$$

$$E_{AttnCAM} = \text{norm}(\text{resize}(\hat{E}_{AttnCAM})_{0,1:})$$

where $h$ denotes the number of heads in the self-attention mechanism.

**Raw Attention**: This baseline considers the attention maps from the last layer, focusing on the weights associated with the [CLS] token. The attention heads are averaged and the resulting explainability map is normalized. The Raw Attention map $E_{Attn}$ is formalized as:

$$\hat{E}_{Attn} = \mathbf{A}^L_{:,0,1:} \in \mathbb{R}^{h \times 1 \times n}$$

$$E_{Attn} = \text{norm}(\text{resize}(\frac{1}{h} \sum_{k=1}^h (\hat{E}_{Attn})_k)) \quad \in \mathbb{R}^{W \times H} \tag{12}$$

These baselines provide a comprehensive set of comparative measures to evaluate the efficacy of our proposed method in the context of explainability for ViTs.

### B.3 DETAILS ON ADAPTING GRADCAM TO VIT

As GradCAM was designed for CNNs without [CLS] tokens, we tried both alternatives, *i.e.* including/excluding [CLS] token, Tab. 7 showcases a comparison of including/excluding the [CLS] token in the gradient computation on ViT-B. We observe that including the [CLS] token result in a marginal improvement. Overall, we would consider both options valid.

GradCAM was originally designed for CNNs, therefore needs some adaptation to work for ViT. In an effort to consolidate the baselines used in this paper, we tried different configurations. One of which is whether or not include the [CLS] token in the gradient computation or not. We note that both alternatives are aligned with the original design of the GradCAM and that this choice is a matter of implementation.

We found that including the [CLS] token was producing better numbers, we therefore used that choice. Indeed, Table 7 shows the results on all the benchmark for GradCAM w/ and w/o the [CLS] token included in the gradient computation and shows that not including it translate in a slight decrease.

Moreover, including the [CLS] token in the gradient computation is the option that makes more sense, as it is the [CLS] that is used to compute the similarity with the text query.

We also tried to use the layer aggregation used in LeGrad for GradCAM and provide an evaluation in Table 7. We apply the same layer aggregation as in LeGrad (see 9) showing a slight improvement on the V7 benchmark. We attribute the only modest improvement to the fact that layers in ViT have different activation value ranges. LeGrad uses only gradients to produce layer-wise explainability maps, thereby avoiding this issue.

### B.4 ADAPTATION OF BASELINE METHODS TO ATTENTIONAL POOLER

In the main manuscript, we introduced our novel method, LeGrad, and its application to Vision Transformers (ViTs) with attentional poolers. Here, we provide supplementary details on how LeGrad and other baseline methods were adapted ViTs employing attentional poolers:

**CheferCAM:** Following the original paper (Chefer et al., 2021) that introduces CheferCAM, we considered the attentional pooler as an instance of a *"decoder transformer"* and applied the relevancy update rules described in equation (10) of that paper (Chefer et al., 2021), (following the

| | mIoU(V7) | Neg ↑(INet) | Pos ↓(INet) |
|---|---|---|---|
| GradCAM w/ [CLS] | 8.72 | 45.26 | 22.86 |
| GradCAM w/o [CLS] | 8.18 | 41.29 | 24.20 |
| Multi-layer w/ [CLS] | 9.51 | 45.31 | 22.54 |
| *LeGrad* | *48.38* | *52.27* | *13.97* |

Table 7: Evaluation of GradCAM with and without [CLS] token as well as with layer aggregation as proposed in eqaution 9 on ViT-B/16 for open-vocabulary detection(V7) and perturbation (ImageNet).

example of DETR). Since the attentional pooler has no skip connection we adapted the relevancy update rule not to consider the skip connection.

**AttentionCAM:** For AttentionCAM, instead of using the attention maps of the last layer, we use the attention maps of the attentional pooler. We found this variant to work better.

**Raw Attention:** Similarly, the Raw Attention baseline was adjusted by substituting the attention maps from the last self-attention layer with those from the attentional pooler.

**Other Baselines:** For the remaining baseline methods, no alterations were necessary. These methods were inherently compatible with the attentional pooler, and thus could be applied directly without any further adaptation.

The adaptations described above ensure that each baseline method is appropriately tailored to the ViTs with attentional poolers, allowing for a fair comparison with our proposed LeGrad method.

### B.5 MITIGATION OF SENSITIVITY TO IRRELEVANT REGIONS:

We observe that for all evaluated explainability methods, SigLIP displays high activations in image regions corresponding to the background. These activations appeared to be invariant to the input, regardless of the gradient computation's basis, these regions were consistently highlighted. To address this issue, we computed the explainability map for a non-informative prompt, specifically `"a photo of"`. We then leveraged this map to suppress the irrelevant activations.

Namely, for an activation $s$ under examination, we nullify any location where the activation exceeds a predefined threshold (set at 0.8) in the map generated for the dummy prompt. Formally, let $E^s$ denote the explainability map for activation $s$, and $E^{empty}$ represent the map for the prompt `"a photo of"`. The correction procedure is defined as follows:

$$E^s_{E^{empty}>th} = 0, \qquad (13)$$

where $th = 0.8$. This method effectively addresses the issue without resorting to external data regarding the image content.

## C ADDITIONAL RESULTS

### C.1 IMAGE CLASSIFICATION

In this section, we extend our evaluation of the proposed LeGrad method to Vision Transformers (ViTs) that have been trained on the ImageNet dataset for the task of image classification. The results of this evaluation are presented in Table 8, also providing a comparison with other state-of-the-art explainability methods.

It shows that LeGrad achieves superior performance on the perturbation-based benchmark, particularly in scenarios involving positive perturbations.

Another observation is that even elementary explainability approaches, such as examining the raw attention maps from the final attention layer of the ViT, demonstrate competitive results. In fact, these basic methods surpass more complex ones like GradCAM (achieving an AUC of 53.1 versus 43.0 for negative perturbations).

| Method | Negative | | Positive | |
|---|---|---|---|---|
| | Predicted ↑ | Target ↑ | Predicted ↓ | Target ↓ |
| rollout (Abnar & Zuidema, 2020) | 53.10 | 53.10 | 20.06 | 20.06 |
| Raw attention | 45.55 | 45.55 | 24.01 | 24.01 |
| GradCAM (Selvaraju et al., 2017) | 43.17 | 42.97 | 26.89 | 26.99 |
| AttentionCAM (Chefer et al., 2021) | 41.53 | 42.03 | 33.54 | 34.05 |
| Trans. Attrib. (Chefer et al., 2020) | 54.19 | 55.09 | 17.01 | 16.36 |
| Partial-LRP (Voita et al., 2019) | 50.28 | 50.29 | 19.82 | 19.80 |
| CheferCAM (Chefer et al., 2021) | 54.68 | 55.70 | 17.30 | 16.75 |
| **LeGrad** | **54.72** | **56.43** | **15.20** | **14.13** |

Table 8: **SOTA comparison on ViT-B/16:** Comparison of explainability methods on perturbation-based tasks on ImageNet-val for a ViT trained on ImageNet.

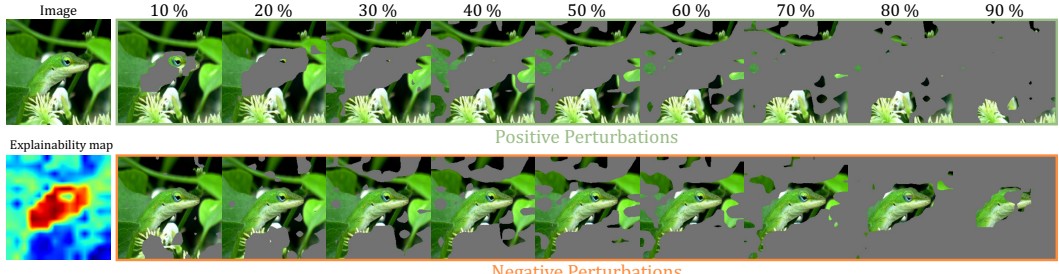

Figure 8: **Example of positive/negative perturbations:** illustration of positive and negative perturbations used in the perturbation-based benchmark. *(Top row)*: positive perturbation. *Bottom row*: negative perturbations

## C.2 PERTUBATION EXAMPLE

Figure 8 illustrates the perturbation-based benchmark of Section 4.1 in the main paper. Given the explainability map generated by the explainability method, for the negative (respectively positive) we progressively remove the most important (respectively the least important) part of the image. We then look at the decline in the model accuracy.

## C.3 QUALITATIVE EXAMPLES

Finally, Figure 11 provides additional qualitative comparisons with state-of-the-art explainability methods, illustrating the efficacy of the proposed approach.

### C.3.1 ABLATIONS OF RELU AND LAYER AGGREGATION

We finally scrutinize the design choices underpinning LeGrad in Table 9. Specifically, we investigate the effect of discarding negative gradients before aggregating layer-specific explainability maps vai ReLU, as well as the implications of leveraging intermediate feature tokens $Z^l$ to compute gradients for each respective layer. We use the framework of the perturbation benchmark delineated in Section **??** and both ViT-L/14 and ViT-H/14 models. The results indicate that the omission of either component induces decline in performance, thereby affirming the role these elements play in the architecture of the method.

## D GRADIENT DISTRIBUTION OVER LAYERS

Figures 9 & 10 extend the "gradient distribution over Layers" analysis conducted Section 4.7 to more backbone size and more set of weights .

|  |  | L/14 | | H/14 | |
| --- | --- | --- | --- | --- | --- |
| ReLU | All Layers | Negative↑ | Positive↓ | Negative↑ | Positive↓ |
| ✗ | ✗ | 47.81 | 20.80 | 57.57 | 21.73 |
| ✓ | ✗ | 49.32 | 19.95 | 59.55 | 19.50 |
| ✗ | ✓ | 52.01 | 16.80 | 60.28 | 18.26 |
| ✓ | ✓ | **54.48** | **15.23** | **61.72** | **18.26** |

Table 9: **Ablation study:** *"ReLU"* corresponds to whether or not negative gradients are set to 0. *"All layers"* corresponds to whether or not the intermediate tokens are used to compute the gradient for every layer or if only the features from the last layer are used. Numbers are the AUC score for the perturbation base benchmark using the target class to compute the explainability map.

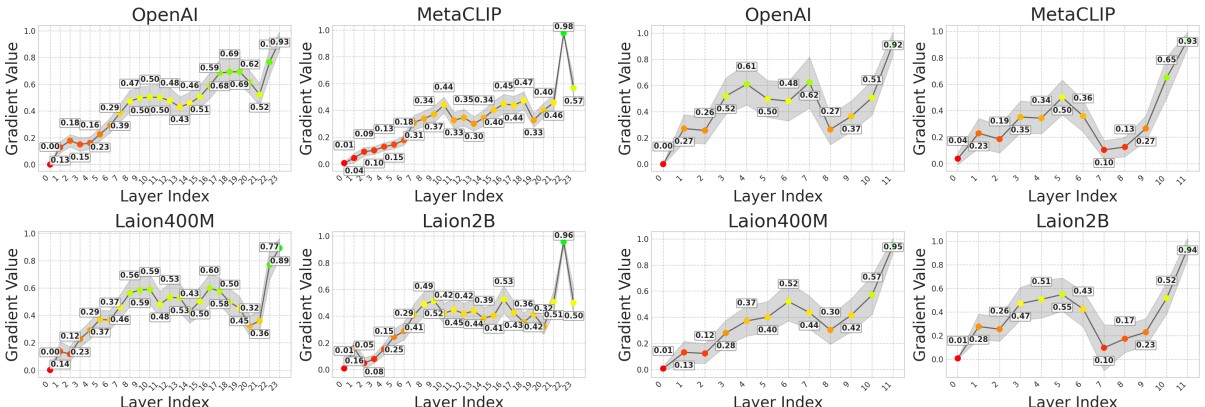

Figure 9: Gradient distribution over layers on PascalVOC for different pretrained ViT-L/14.

Figure 10: Gradient distribution over layers on PascalVOC for different pretrained ViT-B/16.

In Figures 12 & 13 & 14 & 15 & 16 & 17 & 18 provide the gradient distribution over layer independetly for each class in PascalVOC. We observe that for most models and classes, the layers contributing the most were typically located towards the end of the ViT. An interesting exception to this trend was observed for the 'person' class, which exhibited a higher sensitivity to the middle layers across all model sizes and weight sets. We hypothesize that this is due to the high frequency of the 'person' class in the training data, enabling the ViT to identify the object early in the layer sequence, thereby triggering the activation in the middle layers.

Furthermore, we note that the most activated layer varied significantly depending on the class and the model. This variability was observed even between two models of the same size, such as ViT-L(openai) and ViT-L(metaclip). This observation underscores the rationale behind the LeGrad method's approach of utilizing multiple layers, hence alleviating the need to select a specific layer, as the optimal choice would differ from model to model.

| Method | CSDC | PC | DC | D | SD | TS |
| --- | --- | --- | --- | --- | --- | --- |
| GradCAM | 0.75 | 0.67 | 0.68 | 0.91 | 0.7 | 0.48 |
| Rollout | 0.86 | 0.8 | 0.82 | 0.8 | 0.76 | 0. |
| Chefer LRP | **0.91** | **0.92** | 0.89 | 0.9 | 0.74 | 0.95 |
| LeGrad | 0.90 | 0.83 | **0.92** | **1.** | **0.77** | **0.97** |

Table 10: Evaluation of ViT-B/16 on the FunnyBirds Co-12 setup

# E  SANITY CHECK

The Co-12 recipe(Nauta et al., 2023) is a set of 12 properties for evaluating the explanation quality of explainability method for machine learing models. These properties provide a comprehensive framework for assessing how well one can explain the decision-making process of a model. In Hesse et al. (2023) proposed a dataset, called FunnyBirds, to evaluate explainability methods for

visual models. As a sanity check for the proposed LeGrad method, we follow the authors guidelines and evaluate on the provided ViT-B/16[1] using LeGrad, showing improvement over gradient-based methods while being on par with LRP.

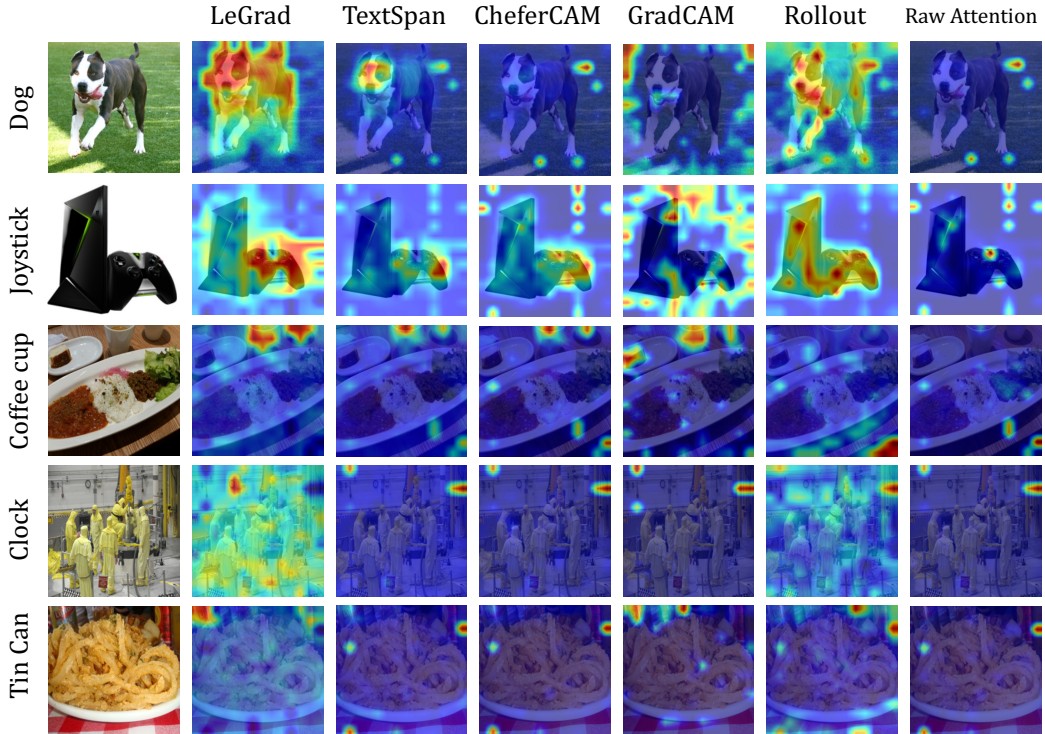

Figure 11: **Qualitative Comparison to SOTA:** visual comparison of different explainability methods on images from OpenImagesV7

## F  PERSONAL AND HUMAN SUBJECTS DATA

We acknowledge the use of datasets such as ImageNet and OpenImagesV7, which contain images sourced from the internet, potentially without the consent of the individuals depicted. We recognize that the VL models used in this study were trained on the LAION-2B dataset, which may include sensitive content. We emphasize the importance of ethical considerations in the use of such data.

---

[1] https://github.com/visinf/funnybirds-framework

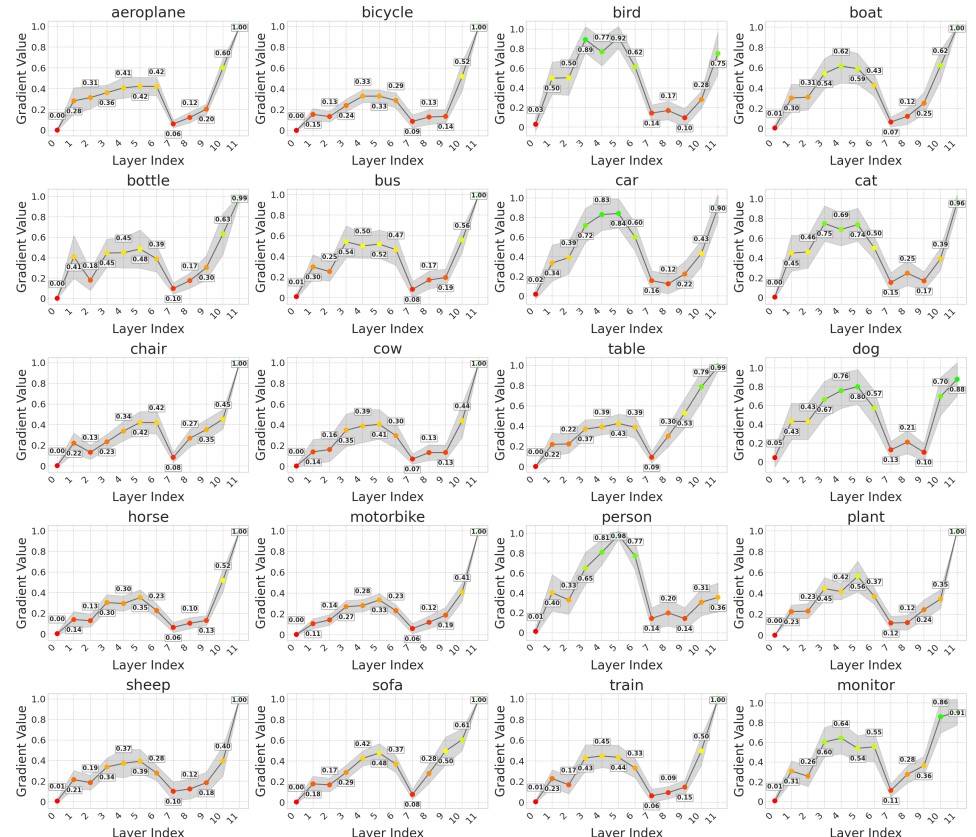

Figure 12: Gradient Distribution over Layers for different classes dataset for Laion2B-ViT-B/16.

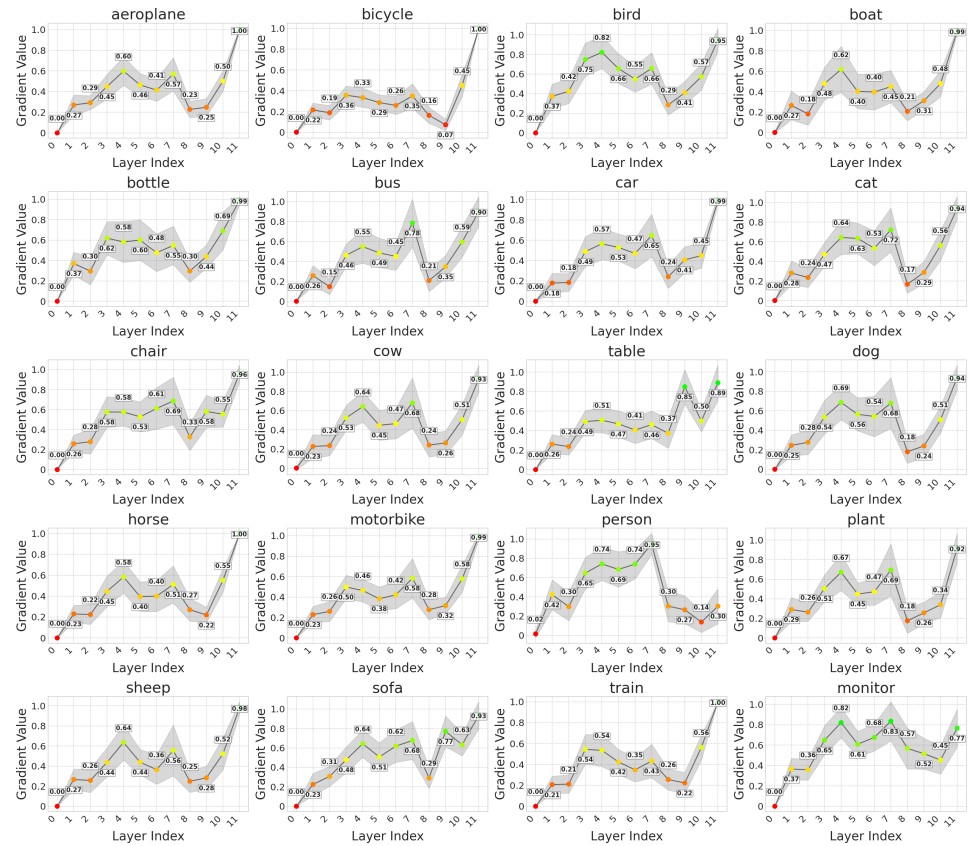

Figure 13: Gradient Distribution over Layers for different classes dataset for OpenAI-ViT-B/16.

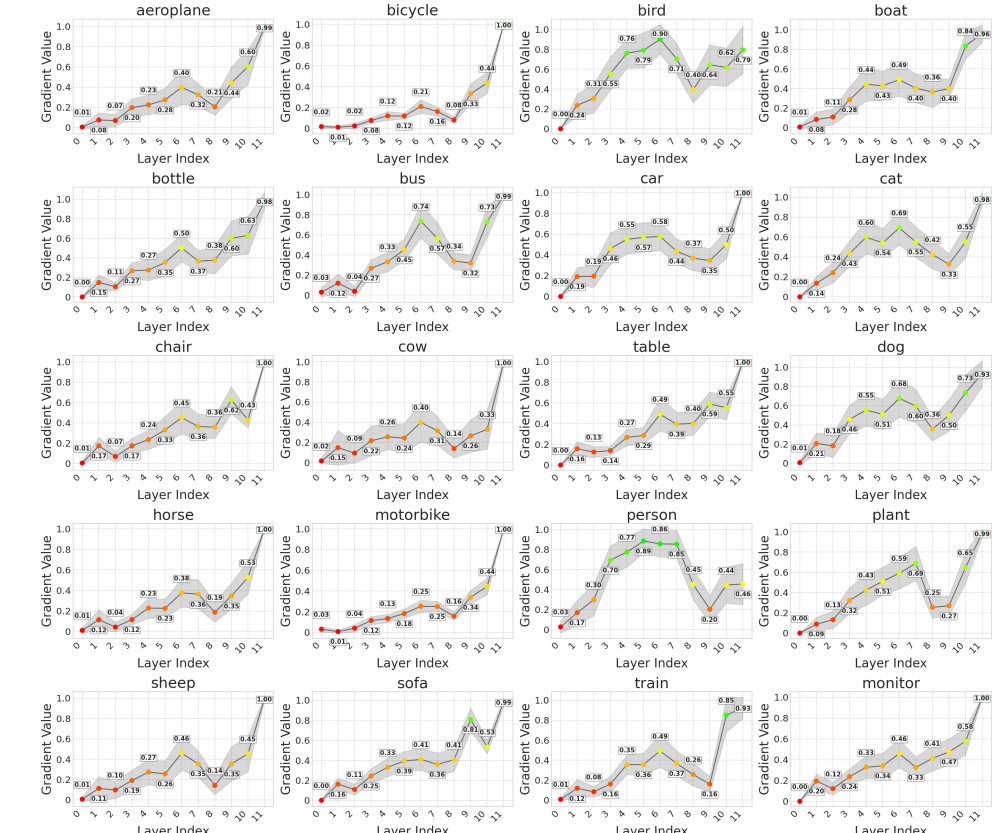

Figure 14: Gradient Distributlion over Layers for different classes dataset for Laion400M-ViT-B/16.

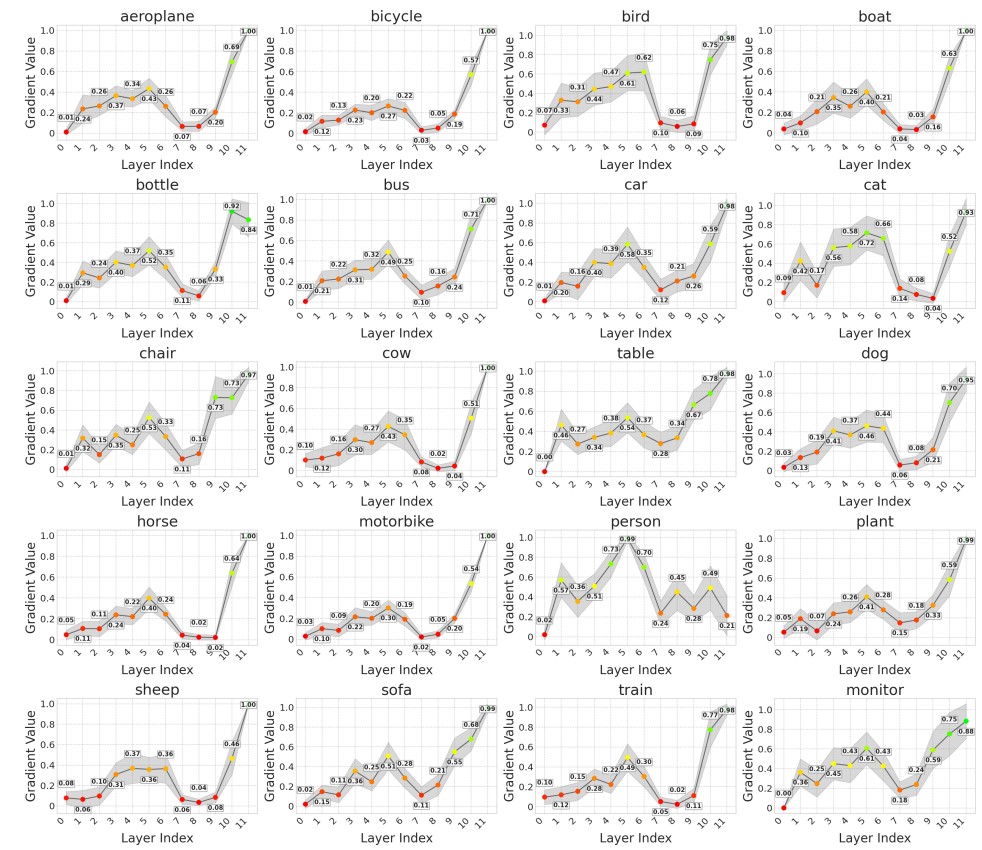

Figure 15: Gradient Distributlion over Layers for different classes dataset for MetaCLIP-ViT-B/16.

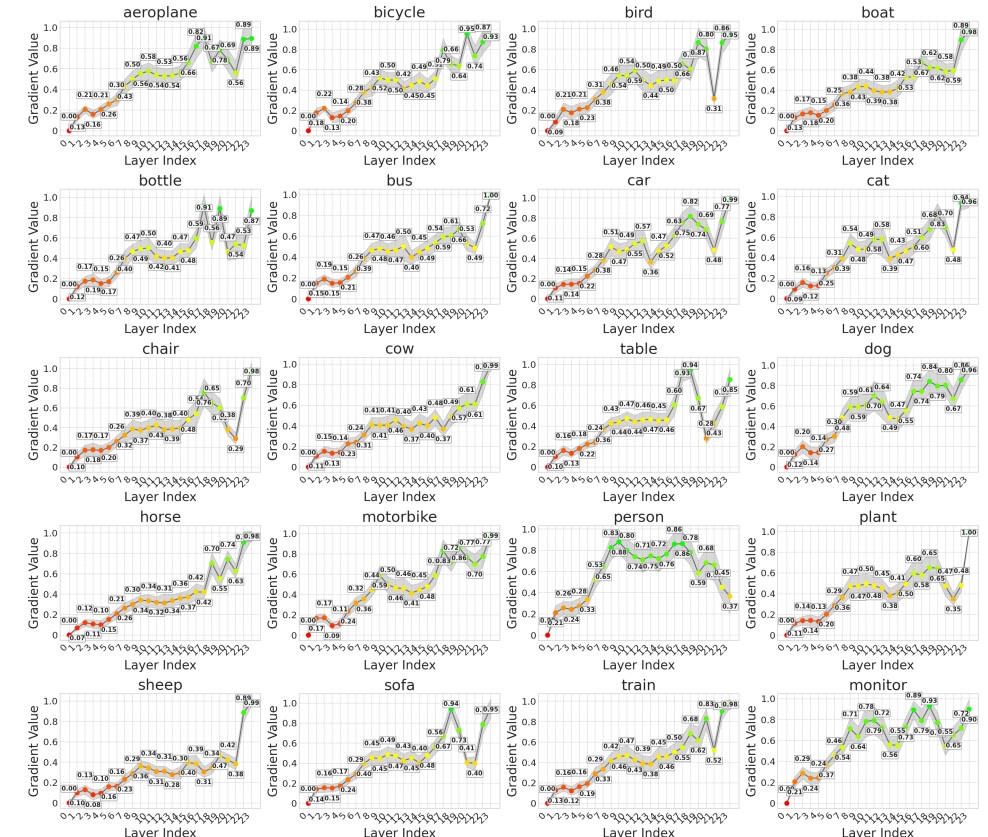

Figure 16: Gradient Distributlion over Layers for different classes dataset for OpenAI-ViT-L/14.

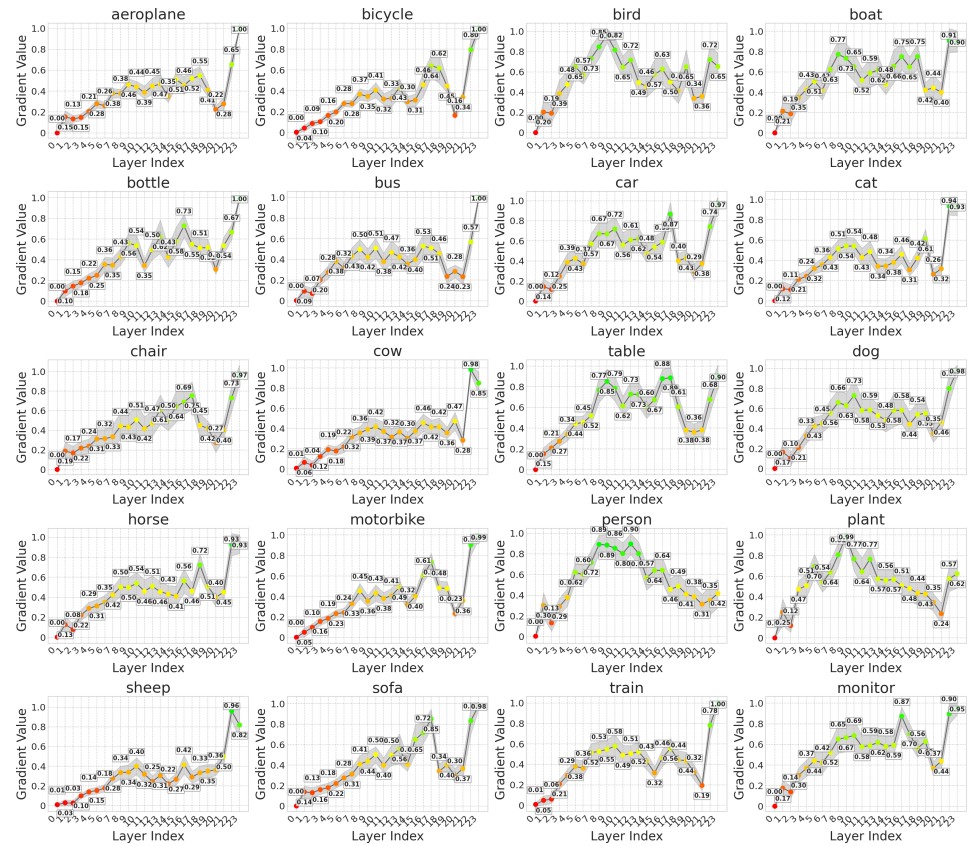

Figure 17: Gradient Distributlion over Layers for different classes dataset for Laion400M-ViT-L/14.

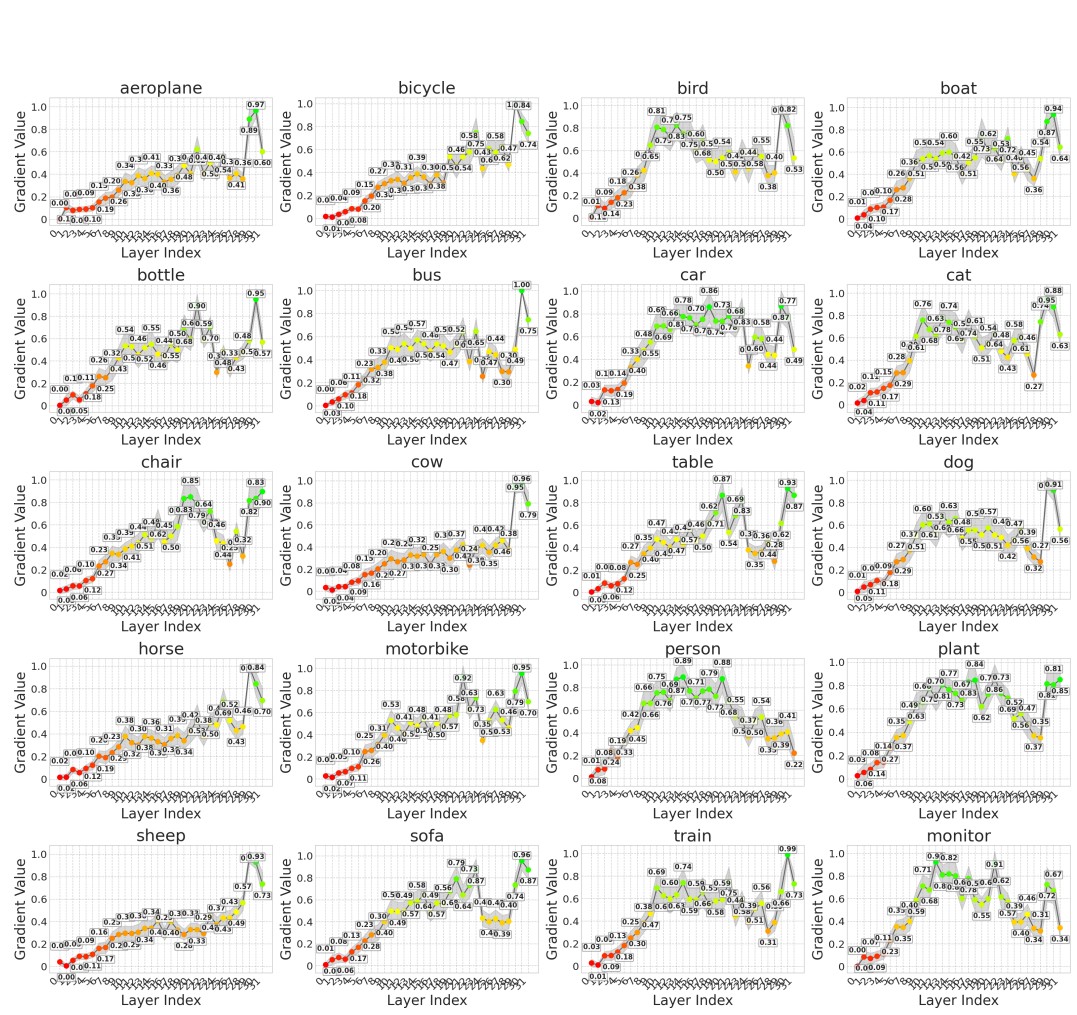

Figure 18: Gradient Distribution over Layers for different classes dataset for Laion2B-ViT-H/14.

