# OpenReview forum: "LeGrad: An Explainability Method for Vision Transformers via Feature Formation Sensitivity"
_ICLR.cc/2025/Conference — ICLR 2025 Conference Withdrawn Submission_

### Official Review · Reviewer_rguk · 2024-10-18

**Soundness:** 3
**Presentation:** 1
**Contribution:** 3
**Rating:** 5
**Confidence:** 4

**Summary:**

This paper addresses the problem of explainable AI specifically designed for Vision Transformers. The authors propose using the derivative of the attention layer at each individual layer. They suggest treating each layer independently by applying a classification head that takes a function of the output from each layer and aggregates the impact across all heads and layers.

**Strengths:**

1) The idea of dividing the explainability into a per-layer impact is sound and reasonable, potentially offering better insights into the information flow throughout the entire architecture.
2) The method is tested across a wide range of ViT adaptations and on classic classification as well as open-vocabulary settings, demonstrating its versatility.
3) The method compared in a versatile applications, especially audio to image tracking.

**Weaknesses:**

1) The writing level is not up to standard, and this affects the clarity and coherence of the paper. The lack of a main chain of thought makes it difficult to follow.

1.a) Typos: influcened -> influenced (line 47), explanability -> explainability (line 49)

1.b) English mistakes: .. the explainability *(of)* those architectures remains a challenge (line 55),
       This facilitates its adoption across various applications and architecture*(s)* (line 67)

1.c) Too long sentences, lack of appropriate punctuation (lines 69-72)

2) The research gap the paper aims to address is unclear. The introduction mentions that "the explainability of those architectures remains a challenge" but doesn’t fully explain why. LeGrad is then introduced as differ from CheferCam, but the main advances LeGrad provides over other methods are not clarified. Why are these specific advances? why are they needed and useful?

3) Lack of intuition behind architectural choices of the approach. For example:  Why using min-max normalization? why do you average on heads and layers in Eq. 5 (it is common to average heads, but not necessarily for layers)? why do you average patches together with CLS for the mapping in line 190 (I familiar with classification ViTs working either on CLS or on average of patch tokens but not the combination of both). All of these make it difficult to follow-up works to grasp the innovative delta and extend it. Moreover, the ablation is not cover parts of the method. Therefore, this is not clear what part of the method is mostly boost the performance.

**Questions:**

1) Why are the results of CheferCam are not correlate with the results stated in the paper of Chefer et al.? for the negative they declare a result of 55.04 on the target (also on imagenet-val dataset) which I could'nt find in your table.
2) I didn't understand from the written if the C mapping is learned once only on top of the last layer and applied to all others? or you have L such mappings, one per layer? Moreover, if learned once only on ImageNet, have you used the same one on other analyzed datasets (i.e. audio)?
3) Explainability methods are preferable when they have some interpretability. How one can use your method in order to better understand the performance of ViT? for example - which layers/heads contributed the most? how can we use this method in order to debug misclassified samples (e.g. in case of classification).

Overall I think that the direction of analyzing each layer separately is a good one, with a potential to improve explainability and even interpretability. Although I think that the paper is lack of intuitions behind some choices, make it difficult for follow-up works. Moreover, the writing level of the paper is somewhat low and hard to follow.

---

### Official Review · Reviewer_oxTL · 2024-10-28

**Soundness:** 2
**Presentation:** 3
**Contribution:** 2
**Rating:** 3
**Confidence:** 4

**Summary:**

This manuscript proposes LeGrad, a saliency method that visualizes the inner behavior of a vision transformer. After obtaining average over output tokens and mapping layer, LeGrad computes gradient of score with respect to attention map. The accumulation of gradients now becomes the saliency map that LeGrad provides.

**Strengths:**

Empirical results exhibit improved quality of saliency, which is validated across numerous benchmarks. I would also like to mention that detailed technical description such as how Grad-CAM can be implemented in ViT with respect to token axis may be valuable to several practitioners.

**Weaknesses:**

Honestly speaking, I think that this manuscript provides not much significant advancement from existing variants of Grad-CAM or existing visualization methods used in ResNets. The summation over gradient with respect to score seems a minor variant of Grad-CAM and highly similar to those existing researches. I would say the novelty of the proposed method lies in detailed engineering such as computation over the token axis and reshaping operations to be compatible with ViTs. A new idea would be something like obtaining saliency maps over multiple layers and averaging them; though it may yield improved quality, it is essentially engineering, not the scientific finding. In summary, I think that LeGrad exhibits little advancement over existing saliency methods, and the proposed method seems mere engineering, not a scientific finding. I wonder whether the LeGrad exhibits improved theoretical properties or was just validated through empirical experiments.

Although the mapping model C can be easily implemented for vision-language models like CLIP, the standard ViT requires the mapping model C through additional classifier layers that are fine-tuned after appending them to ViT. In other words, the proposed method cannot be immediately applied to standard ViT and requires additional fine-tuning, which makes it difficult to be deployed immediately. Indeed, ViT with GAP is rarely used; standard ViT uses an output score obtained from the class token without average operation. This issue also raises limitations for the proposed method.

Minor comments: LayerNorm is omitted in Eq. 1.

**Questions:**

See the weaknesses.

---

### Official Review · Reviewer_Vs78 · 2024-11-01

**Soundness:** 3
**Presentation:** 2
**Contribution:** 2
**Rating:** 5
**Confidence:** 5

**Summary:**

This paper studies the explainability of Vision Transformers (ViTs). The authors propose a new gradient based method, LeGrad, to produce explanation maps given pre-trained ViTs and specific inputs. LeGrad aggregates the gradient signal and combines the activations of the last and intermediate tokens to produce the merged explanation maps. Experiments show that the proposed LeGrad achieves better results in segmentation, perturbation, and open-vocabulary settings.

**Strengths:**

1. Unlike previous works, this paper extends the scope of ViT explainability to include new models like OpenCLIP ViTs and new settings like open vocabulary.

2. this paper is well-written and easy to follow.

3. comprehensive experiments. Tables 1 and 2 show that LeGrad significantly outperforms the baseline methods.

**Weaknesses:**

1. the technical contribution of LeGrad is not very significant. The idea of using gradient as an explanation signal and aggregating information across layers [1, 2] has been well-studied and widely applied in the literature on ViT explainability, making this work incremental.

2. The method sums up the gradients across layers instead of performing matrix multiplication. This is different from the attention flow model in [1, 2, 3]. A detailed analysis of the motivation for summing up the signals would largely strengthen this paper.

3. Similar to weakness 2. The proposed LeGrad method mainly focuses on the gradient itself as an explanation signal instead of the activation. However, the reason behind this design has not been clearly explained.
[1] Generic Attention-model Explainability for Interpreting Bi-Modal and Encoder-Decoder Transformers. ICCV 2021.
[2] Transformer Interpretability Beyond Attention Visualization. CVPR 2021.

**Questions:**

The visualization results (Figures 1, 5, 7, 11) indicate that LeGrad may produce a noisy background in the explanation maps, a problem studied in previous work [1, 2]. Is there any analysis on this issue? A possible reason is that the gradient signal is noisy in background areas and thus needs calibration. Can the vector norm methods in [1, 2] help improve the clearness in the background produced by LeGrad?

[1] Token Transformation Matters: Towards Faithful Post-hoc Explanation for Vision Transformer. CVPR
2024.

[2] Attention is Not Only a Weight: Analyzing Transformers with Vector Norms. EMNLP 2020.

---

### Official Review · Reviewer_Tim1 · 2024-11-08

**Soundness:** 3
**Presentation:** 3
**Contribution:** 2
**Rating:** 5
**Confidence:** 3

**Summary:**

This paper introduces LeGrad, a new explainability method tailored for Vision Transformers (ViTs), which leverages gradients with respect to attention maps to generate relevancy maps that highlight the most influential parts of an input image for model predictions. By conducting a layer-wise analysis, LeGrad aggregates information across multiple layers, revealing the contributions of individual layers to the model's decision-making process. The method is evaluated across various tasks, including object segmentation and open-vocabulary detection, demonstrating good performance and more focused visual explanations compared to existing state-of-the-art methods like GradCAM and CheferCAM.

**Strengths:**

- LeGrad effectively utilizes a layer-wise approach to explainability, allowing it to aggregate gradients from multiple layers of the Vision Transformer. This comprehensive analysis provides a more nuanced understanding of how different layers contribute to the model's predictions, enhancing interpretability.

- By focusing on gradients with respect to attention maps, LeGrad captures the sensitivity of feature representations, enabling the generation of relevancy maps that highlight the most influential parts of an image.

- LeGrad demonstrates strong performance across various challenging tasks, such as object segmentation and open-vocabulary detection, achieving comparable results compared to other state-of-the-art explainability methods.

**Weaknesses:**

- Missing references [1,2,3].

- The idea of the proposed method sounds very normal. It is very common and has been well studied to use gradient and attention to localize the object in Transformer-based architecture. Talking about text-image alignment, [4] also shows a more precise segmentation results than the proposed method.

- Figure 4 shows typo in the x-axis. The x-axis means the "% of accumulated layer", so it should be 10, 20, ..., 100 rather than 0.1, 0.2, ..., 1.0.



References:

[1] IA-RED^2: Interpretability-Aware Redundancy Reduction for Vision Transformers.

[2]  Emerging properties in self-supervised vision transformers.

[3] Exploring Visual Explanations for Contrastive Language-Image Pre-training.

[4] Open-Vocabulary Panoptic Segmentation with Text-to-Image Diffusion Models

**Questions:**

NA

---

### Note · Authors · 2024-11-15

I have read and agree with the venue's withdrawal policy on behalf of myself and my co-authors.